# *Nigella sativa*-Floral Honey and Multi-Floral Honey versus *Nigella sativa* Oil against Testicular Degeneration Rat Model: The Possible Protective Mechanisms

**DOI:** 10.3390/nu15071693

**Published:** 2023-03-30

**Authors:** Mona S. Almujaydil, Reham M. Algheshairy, Raghad M. Alhomaid, Hend F. Alharbi, Hoda A. Ali

**Affiliations:** 1Department of Food Science and Human Nutrition, College of Agriculture and Veterinary Medicine, Qassim University, Buraydah 51452, Saudi Arabia; m.almujaydil@qu.edu.sa (M.S.A.); r.alhomaid@qu.edu.sa (R.M.A.); hf.alharbi@qu.edu.sa (H.F.A.);; 2Department of Nutrition and Clinical Nutrition, College of Veterinary Medicine, Cairo University, Cairo 11562, Egypt

**Keywords:** testicular degeneration, *Nigella sativa* floral honey, multi-floral honey, *Nigella sativa* oil, pituitary–testicular axis, apoptosis, redox status

## Abstract

The male reproductive function, particularly the testes, and the related hormones are sensitive to various xenobiotics. This work aimed for the first time to assess *Nigella sativa* floral honey (NS floral honey) and multi-floral honey (M-floral honey) versus *Nigella sativa* oil (NS oil) against rat testicular degeneration induced with azathioprine (AZA). A total of 40 male Wister rats were assigned into 5 groups: (1) control, (2) 15 mg/kg of AZA, (3) AZA + 1.4 mL/kg of M-floral honey, (4) AZA + 1.4 mL/kg of NS floral honey, and (5) AZA + 500 mg/kg of NA oil. Total testosterone (TT), free testosterone (FT), free androgen index (FAI), gonadotrophins, sex-hormone-binding globulin (SHBG), apoptosis markers, and redox status were assessed to clarify the possible protective mechanisms. Pituitary–testicular axis disruption, apoptosis markers, poor redox status, and sperm quality (count, viability, and motility) were set with AZA. Serum TT, SHBG, and absolute and relative testis weight were significantly restored in the NS oil and NS floral honey groups. Meanwhile, the NS oil group exhibited a significant elevation in FT and FAI. Serum gonadotrophins increased significantly in the NS floral honey (*p* < 0.01) and M-floral honey and NS oil (*p* < 0.05) groups. Testicular caspase-3, caspase-9, and nitric oxide showed significant improvement in the NS floral honey and NS oil groups. NS oil supplementation significantly normalized redox status (*p* < 0.05), whereas NS floral honey improved malondialdehyde and superoxide dismutase activity. Sperm quality exhibited a significant improvement in the NS oil group (*p* < 0.05). M-floral honey did not show reliable results. Although NS floral honey could protect against testicular damage, it did not upgrade to the level of improvement achieved with NS oil. We claim that further clinical studies are essential for focusing on the quality and quantity of bioactive constituents.

## 1. Introduction

Infertility is one of the major problems among young couples, such that the percentage of infertile men ranges from 2.5% to 12% [1]. The gonad is one of the main target organs for xenobiotics including pollution, drugs, and toxins that might contaminate food, drinking water, and the air or those from the materials used in buildings and industries [2,3]. According to a recent study, exposure to these toxins is linked to public health risks that can alter the endocrine system, disturb hormonal balance, and negatively impact fertility [4].

Despite the use of modern therapies to support sexual desire and relieve sexual dysfunction, natural alternative medicine has earned substantial consideration among scientists due to its wide safety margin and low price [5]. Among those natural medicines are botanical sources, such as *Nigella sativa* [6] and honey [7]. *Nigella sativa* Linn., a member of the Ranunculaceae family and commonly referred to as “black seed”, or Habbatul Barakah, has long been utilized in the Middle East, Northern Africa, and Asia as a traditional medicine for digestive and respiratory disorders and boosts the immune system. [8]. Many bioactive substances, including nigellicine, nigellimine, nigellidine, alpha-hederin, thymoquinone, thymohydroquinone, and dithymoquinone, are present in *Nigella sativa* [9]. Pharmacologically, the seeds of *Nigella sativa* have antioxidant, anti-inflammatory, antiviral, antibacterial, and antidiabetic effects [10,11]. Moreover, *Nigella sativa* oil (NS oil) has a favorable impact on testicular protective properties, particularly, semen characteristics and testosterone [6,12].

Honey is a naturally occurring bee product made from nectar that is gathered from flowers. In addition to its nutritional value, it possesses significant health benefits [13]. It has biologically active compounds such as phenolic acids and flavonoids [14], antioxidants [3], and a wide range of biological properties [15]. Studies have shown that honey revitalizes spermatogenesis in mice exposed to environmental contamination [16].

*Nigella sativa* floral honey (NS floral honey) is a mono-floral honey that is found in beehives located near *Nigella sativa* fields so that the bees obtain nectar predominantly from *Nigella sativa* flowers. Meanwhile, multi-floral honey (M-floral honey) is of a non-specific, rather than black seed, flower origin, i.e., it originates from a mixed blossom and is commonly sold commercially.

Despite many types of research that have studied the impact of the Nigella sativa plant and honey on the male reproductive system and fertility, their exact role is still confounding and needs a more comprehensive view. In addition, NS floral honey has not yet been explored as a natural product to attenuate the cases of testicular dysfunction. The present study hypothesized that NS floral honey carries some bioactive compounds likely found in *Nigella sativa*, suggesting that it has a dual therapeutic effect. Therefore, the present study for the first time aimed to explore the effects of NS floral honey and M-floral honey versus NS oil to modulate testicular dysfunction and the associated reproductive hormones in an azathioprine (AZA)-induced testicular degeneration rat model, highlighting the possible protective mechanisms via pituitary–testicular axis hormones, redox balance, and germ cell apoptosis.

## 2. Materials and Methods

### 2.1. Ethical Standard

According to the International Animal Ethics Committee, the current study received approval from the Committee of Health Research Ethics at Qassim University in the Kingdom of Saudi Arabia under the number “23-24-23”.

### 2.2. Animals

Adult healthy male Wistar rats weighing approximately 160 ± 10 gm were obtained from Riyadh, the University of King Saud laboratory center, KSA. The animals were reared in cages in an animal-specific room at the College of Agriculture and Veterinary Medicine at Qassim University in Saudi Arabia. The room had a photoperiod of 12 h light and then 12 h dark at a temperature of 23 ± 2 °C. The rats were acclimatized for one week, during which they were offered water and a commercial diet till appetite. The commercial diet was obtained from “The Wafi company for animal feed, Qassim, KSA”. The diet is formulated to furnish the nutrient requirement for lab animals, as recommended by the requirements of the National Research Council [17]. General guidelines for using and caring for animals for scientific purposes were considered. The Deanship of Scientific Research, Qassim University, suggested certain animal care standards and ethics, which were followed in the present experiment.

### 2.3. Nigella Sativa Oil

*Nigella sativa* oil was purchased from (Zamzam Company International for natural oils, herbs, and cosmetics, London, UK) under the Ministry of Health, Cairo, Egypt license. *Nigella Sativa* seeds were only squeezed once on the cold press base to ensure 100% premium quality *Nigella Sativa* oil with no additives and no preservatives.

### 2.4. Honey

Two varieties of honey were used: the first was mono-floral honey (*Nigella sativa* floral honey and NS floral honey) for which the bees obtain nectar from *Nigella sativa* flowers, i.e., these constitute the predominant portion of the honey. The second one was a multi-floral honey (M-floral honey), which is of a non-specific rather than black seed, flower origin, i.e., a mixed blossom. NS floral honey was obtained from (Honey Garden Mountain Company, Riyad, Saudi Arabia), whereas M-floral honey was obtained from the commercial honey market, Riyad, KSA.

### 2.5. Testicular Degeneration Model

A commercial formulation of azathioprine, AZA (Imuran) as a 50 mg tablet (EXCELLA Gmbh & Co., Aspen Pharma Trading Limited, Citywest, Dublin, Ireland KG, Germany), was chosen with a daily oral dosage of 15 mg/kg body weight in a saline solution for 4 weeks to induce the testicular degeneration rat model [18,19].

### 2.6. Experimental Protocol and Sampling

Forty rats were divided into five groups at random. (*n* = eight). All the animals in each group were weighed before the experiment began. Group (1): the negative control healthy group received a saline solution orally using a gastric gavage tube. Group (2): the positive control group received the same as group 1 for 2 weeks. Groups 3, 4, and 5: the treatment groups received 1.4 mL/kg body weight of M-floral honey, 1.4 mL/kg body weight of NS-floral honey, and 500 mg/kg body weight of NS oil [20] respectively for 2 weeks. After 2 weeks, groups (2, 3, 4, and 5) were subjected to 15 mg/kg body weight AZA treatment orally in a saline solution daily for 4 weeks. The animals in the treatment groups were concomitantly administered both AZA and the supplements for 4 weeks, i.e., the actual duration of the experiment was 6 weeks. The two kinds of honey were freshly prepared using distilled water as a vehicle. The doses of AZA and all the supplements used in the present study were calculated according to the corresponding dose or consumption for humans following the conversion equation previously suggested in [21]. Supposing that the therapeutic dose of AZA for humans is 1–4 mg/kg, a human consumes about 1 spoonful of honey per day and 1 teaspoonful of NS oil per day. Fortunately, the calculated doses corresponded to some previous studies.
Animal dose (mg/kg) = HED (mg/kg) × CF
where HED = human equivalent dose and CF = conversion factor (6.17 for a rat).

The rats were weighed weekly to calculate the exact dosage of all the supplements. At the end of the experiment, the animals were weighed, blood was gathered from the rats via the ocular route, and serum was obtained after 10 min of centrifugation at 2000× *g*. The serum was kept in a deep freezer at −20 °C for further hormonal assays. Three animals from each group were anesthetized and sacrificed. The testes with epididymis were excised, and the left testes were used for measuring absolute and relative testis weight, sperm count, and viability. The right testes were cleaned using phosphate-buffered saline (PBS) and rapidly frozen at −20 °C to determine antioxidant activities (redox balance), fructose and nitric acid (NO) concentrations, apoptotic enzyme activities (caspase-3 and caspase-9), as well as for histopathological examination.

### 2.7. Relative Testis Weight

The left testes were washed in a normal saline solution and weighed using a sensitive balance, and then we recorded the absolute testis weight. The relative testis weight was calculated as the ratio of the testis wet weight (g) to body weight (g).

Relative testis weight = Left testis/body weight × 100 [22].

### 2.8. Hormone Assay

The serum total testosterone (TT) concentration was measured using the ELISA, (MONOCENT, Inc., Los Angeles, CA, USA) ref. NO 1-1263. The intra-assay and inter-assay CVs were <3.8% and <6.7%, respectively. The assay range and sensitivity were 3–9.5 ng/mL and 1.16%, respectively. Sex-hormone-binding globulin (SHBG) was previously named androgen-binding globulin (ABG) (SUNLONGBIOTETECH, Co., Ltd., HangZhou, China), cat. no., SL1576 Hu. The intra-assay and inter-assay had CVs of <10.5% and <12.3%, respectively. The assay range was 0.3–20 ng/mL, and the sensitivity was 0.05 ng/mL. The free testosterone (FT) was obtained using substrate TT from SHBG, and then the TT/FT ratio was calculated. The calculation of the free androgen index (FAI) was carried out following the equation suggested by [23].
FAI = total testosterone/SHBG × 100

The concentration of gametogenic hormone (FSH) was determined using a commercial ELISA kit (Elabscience, Houston, TX, USA), cat. no: E-EL-R0391. The intra-assay and inter-assay CVs were <8.1% and 8.9%, respectively. The assay range was 3.13–200 ng/mL, and the sensitivity was 1.88 ng/mL. Interstitial-cell-stimulating hormone (ICSH/LH) was estimated using a specific commercial rat ELISA kit (Elabscience, USA), cat. no. E-EL-R0026. The intra-assay and inter-assay CVs were <6.4% and <8.6, respectively. The assay range was 1.56–100 mIU/mL, and the sensitivity was 0.94 mIU/mL. All the hormone assays were carried out in triplicate following the manufacturer’s specifications.

### 2.9. Preparation of Testicular Homogenate

The preparation was performed following the manufacturer’s instructions. Briefly, the homogenization of testis specimens was carried out after the addition of PBS (pH 7.4). The supernatant was collected carefully after centrifuging for 20 min at 1200× *g* at 4 °C. An aliquot of the supernatant was kept for the caspase-3 and caspase-9 assays. The protein content of the supernatant was assayed using [24] Lowry (1951). Caspase-3 assay activity was detected with the ELISA (SUNLONGBIOTETECH, Co., Ltd., China) Cat. No. (SL0152Ra). The intra-assay and inter-assay CVs were <10.6% and 12.4%, respectively. The assay range was 0.06–4 ng/mL, and the sensitivity was 0.01 ng/mL. Caspase-9 activity was assayed with the ELISA (SUNLONGBIOTETECH, Co., Ltd., China), cat. no. SL0154Ra. The intra-assay and inter-assay CVs were < 10.8% and 12.7%, respectively. The assay range was 0.08–4 ng/mL, and the sensitivity was 0.025 ng/mL. The measurements were carried out according to the manufacturer’s instructions. Part of the supernatant was used for the determination of fructose, NO, antioxidant activities, and lipid peroxidation biomarkers. NO was determined using a commercial calorimetric kit (Biodiagnostic, Diagnostic, and Research Reagents, Cairo, Egypt) cat. no. NO 25 33. The measurement was carried out in triplicate following the manufacturer’s specifications. The glutathione peroxidase (GHPx) and superoxide dismutase enzyme (SOD) contents were determined using laboratory kits (Biodiagnostic, Diagnostic, and Research Reagents, Cairo, Egypt), cat. no. GP 25 24 and SD 25 21, respectively. Meanwhile, catalase enzyme (CAT) was assayed using a kit (Elabscience, USA), cat. no. M-BC-K031-S. The MDA was assessed using kits (Biodiagnostic, Diagnostic, and Research Reagents, Cairo, Egypt), cat. no. MD 25 29.

### 2.10. Sperm Count and Viability

The epididymis was carefully removed and placed in a Petri plate containing 1 mL of a normal saline solution at 37 °C. The epididymis was cut into three small finely minced portions (caudal, corpus, and caput) to allow the sperm to swim out. The solution containing the sperms was diluted in a normal saline solution (1:10) and centrifuged at 200× *g* for 3 min. After centrifugation, the supernatant was used for the sperm count and fructose estimation. A small quantity of the prepared epididymal sperm suspensions from each portion of epididymis mentioned previously was separately used for the sperm count after being diluted with 10% neutral formalin (Sigma, Livonia, MI, USA) 1:40. The diluted mixtures were put in a Neubauer chamber (Neubauer’s hemocytometer, Germany) using a Pasteur pipette. A light microscope ×400 was used to count the sperms at the 4 specific corners of the sliding square [25]. Sperm viability was evaluated using a 5% saline solution with Nigrosin–Eosin Y staining (Biodiagnostic, Diagnostic, and Research Reagents, Cairo, Egypt), cat. no. NE 27 27. A cell with an intact membrane does not take up the Eosin Y stain. On a glass slide, 40 μL of fresh sperm solution was added along with 1% eosin. The smear was examined under a light microscope after being allowed to air-dry. After staining, live sperm remained unstained. Each sample had at least 250 sperm counted in 10 fields, and the percentage of viable sperm was confirmed [26]. The fructose was determined with the colorimetric method using a laboratory kit (Biodiagnostic, Diagnostic, and Research Reagents, Cairo, Egypt) cat. no. FR 23 10 at an absorbance of 495 nm.

### 2.11. Histological Examination

The testis specimens were processed as routine with paraffin wax, stored in a 10% neutral buffered formalin aqueous solution, and stained with H&E (Hematoxylin and Eosin) for histological observations [27].

### 2.12. Statistical Analysis

Data values were represented as means of the standard errors. For each measured parameter, a straightforward one-way analysis of variance (ANOVA) test was conducted. The negative control and model animals were compared using post hoc analysis and the Mann–Whitney test. In addition, comparing the model with the supplemented groups (M-floral honey, NS floral honey, and NS oil) with *p* < 0.05 reflects a statistical difference.

## 3. Results

### 3.1. Body Gain and Relative Testis Weight

The model group showed a significant reduction (*p* < 0.05) in testis weight and relative testis weight and a significant reduction in body weight gain at *p* < 0.01 compared with the negative control group. The groups offered either NS floral honey or NS oil exhibited a significant increase in the mentioned parameters (*p* < 0.05) relative to the model group (Table 1). On the other side, the M-floral honey group recorded a significant increase (*p* < 0.05) in the relative testis weight only.

### 3.2. Testosterone, SHBG, and Related Parameters

The model rats had a significant negative effect (*p* < 0.05) on testosterone and the related parameter assays, reflected in a decrease in the serum levels of TT (2.61 ± 0.41 ng/mL), FT (0.78 ± 0.09 ng/mL), and FAI (141.82 ± 4.28); meanwhile, SHBG (1.84 ± 0.18 ng/mL) recorded a significant decrease (*p* < 0.01) compared with the negative control group (Table 2). The administration of NS floral honey caused a significant increase (*p* < 0.05) in serum TT and SHBG comparable to the positive control group. On the other hand, the NS oil group achieved a significant increase (*p* < 0.05) in the serum levels of TT, SHBG, and FAI while recording a highly significant increase (*p* < 0.01) in serum FT compared with the model group. All the experimental treatments did not influence the TT/FT ratio.

### 3.3. Gonadotrophins

The decrease in the serum levels of gonadotrophic hormones (FSH and LH) was significant (*p* < 0.05) in the AZA model group (4.11 ± 0.07 and 6.23 ± 0.49 ng/mL) compared with the negative control group (4.56 ± 0.11 and 8.78 ± 0.56, respectively) (Table 3). The administration of honey from both sources (M-floral honey or NS floral honey) significantly increased the levels of FSH and LH at *p* < 0.05 with an exception in NS floral honey, which increased FSH levels significantly at *p* < 0.01. Likewise, the animals that received NS oil showed a significant increase (*p* < 0.05) in the serum levels of gonadotrophic hormones relative to the positive control group.

### 3.4. Apoptosis Markers

The AZA model rats recorded a significant increase (*p* < 0.05) in caspase-3 (25.63 ± 2.54 ng/mg protein) compared with the negative control group (10.32 ± 1.43 ng/mg protein). Meanwhile, the caspase-9 level was significantly elevated (*p* < 0.01) to 2.83 ± 0.13 ng/mg protein compared with the negative control group, which recorded 1.70 ± 0.17 ng/mg protein. Likewise, a significant increase (*p* < 0.05) in the testicular NO level was recorded in the model group at 0.141 ± 0.014 µmol/mg protein compared with the negative control group at 0.064 ± 0.008 µmol/mg protein (Table 4). Groups that received NS floral honey or NS oil exhibited significant reductions (*p* < 0.05) in testicular caspase-3, caspase-9, and NO with an exception for caspase-9, which recorded a significant decrease (*p* < 0.01) in the group offered NS oil. The administration of M-floral honey achieved significant decreases in caspase-9 (*p* < 0.05) and NO (*p* < 0.01) relative to the model.

### 3.5. Redox Balance (GSH-Px and SOD; CAT and MDH)

The obtained results (Table 5) show that the AZA model group caused an interruption in oxidative stress represented by the significant decrease (*p* < 0.05) in testicular GSH-Px and SOD activities (5.43 ± 1.28 and 11.61 ± 1.2 U/mg protein) and the elevation (*p* < 0.05) in the testicular MDA level (45.7 ± 3.7 nmol/g protein), respectively, compared with the negative control group. The group offered NS floral honey significantly (*p* < 0.05) normalized MDA and SOD, the values of which might be near the normal values recorded in the negative control group. The group that received NS oil achieved a significant (*p* < 0.05) balance of redox status represented by the increases in GSH-Px, SOD, and CAT activities and a decrease in MDA, whereas M-floral honey had no significant influence on redox status.

### 3.6. Sperm Quality

A highly significant reduction (*p* < 0.01) in the caudal epididymal sperm count was recorded in the model group. Meanwhile, the results for live sperm viability% and testicular fructose show a significant reduction and significant increase (*p* < 0.05), respectively, in the model rats compared with the negative control group (Table 6). The group that received NS oil showed a significant increase (*p* < 0.05) in caudal epididymal sperm count and sperm viability%; meanwhile, testicular fructose concentration was significantly decreased (*p* < 0.05) in the model group. Both sources of honey (M-floral honey and NS floral honey) showed a nonsignificant increase in caudal epididymal sperm count and sperm viability% besides a slight decrease in testicular fructose concentration. Epididymal sperm count in the corpus and caput was not affected by any of the experimental treatments.

### 3.7. Histological Examination

Microscopically, the testes of rats from the negative control group showed a typical histological structure composed of seminiferous tubules with normal spermatogonia cells and complete spermatogenesis (Figure 1A,B). On the contrary, the examined sections from the positive control group (AZA model) revealed a marked degeneration and necrosis of the spermatogonia cells lining seminiferous tubules, interstitial edema, and spermatid giant cells in the lumen (Figure 1C,D). However, the testes of rats from the group that received M-floral honey revealed the formation of spermatid giant cells in the lumen of the seminiferous tubule (Figure 1E) and the necrosis of some Leydig cells (Figure 1F). On the other hand, some examined sections from the group offered NS floral honey demonstrated the degeneration and necrosis of some spermatogonia cells lining some seminiferous tubules (Figure 1G), whereas other sections revealed no histopathological alterations (Figure 1H). Otherwise, the testes of rats from the group supplemented with NS oil manifested the standard histological structure of seminiferous tubules (Figure 1J).

## 4. Discussion

Since gonadal toxicity is usually associated with inflammation, oxidative stress, and apoptosis with the interruption of the hormonal pathway, herein, these pathways were traced using the natural products multi-floral honey, *Nigella sativa* flower honey, and *Nigella sativa* oil. Azathioprine (AZA) was chosen in the present study to induce testicular degeneration (model), as previously reported [18].

The AZA model group caused significant reductions in body weight gain, testicular weight, and relative testis weight (similar results have been previously reported in many studies [28,29]), indicating a disruption of spermatogenesis and steroidogenesis [30]. The administration of NS floral honey or NS oil was found to normalize body weight gain, testicular weight, and relative testicular weight, reflecting good functioning of the testes [31,32]. Physiologically, the weight and size of the epididymis and testes increased in the presence of vitamins, zinc, magnesium, and copper, which are components of NS. However, our results contrast with the observations by [33]. The significant increase in the relative testicular weight recorded in the group offered M-floral honey was attributed to the lower body weight gain in that group.

TT is the main hormone responsible for the process of spermatogenesis and the normal function of the testicles [34]. The significant reduction in TT levels shown in the model group is in line with previous work [18,35]. This reduction might be attributed to the hypothalamic–pituitary–adrenal axis [36]. Serum TT level reduction is usually accompanied by an interruption in spermatogenesis and damage to Leydig cells [37] and may be attributed to the oxidative stress induced by this drug [18]. The NS floral honey and NS oil treated groups rebalance the TT to be near the value of the negative control group. These findings are in line with recent studies by many researchers [3,12,38]. Mice given black seed oil had considerably higher serum TT concentrations [39]. Kaliandra honey had a significant effect on improving TT levels [40]. Meanwhile, this observation was not clear with M-floral honey.

SHBG, previously named androgen-binding protein (ABP), binds to androgens [41]. This compound has recently been shown to have a pivotal role in determining fertility and the efficiency of reproductive performance, indicating that SHBG might serve as a sexual activity indicator. Through the hypothalamic–pituitary axis, SHBG can regulate the bioactivity of sex steroids by limiting their diffusion into target organs and increasing total androgen and estrogen concentrations [29]. Likewise, the metabolic clearance rate of sex hormones is regulated by SHBG [42]. The highly significant reduction in SHBG recorded in the AZA model group indicated hypogonadism due to low testosterone and low sperm production [43]. Conversely, the NS floral honey and NS oil groups showed a significant improvement in SHBG, indicating an assurance of maintaining high levels of TT and good Sertoli cells. In rats, it has been established that Sertoli cells generate SHBG, which affects the performance of androgen in the epididymis and seminiferous tubules [44]. In the present study, an increase in SHBG was accompanied by an increase in TT, which keeps the level of free testosterone more constant [45]. SHBG controls the physiological effects of total, free, and/or bioavailable sex steroid concentrations [46]. Interestingly, this is the first time the influence of *Nigella sativa* and honey on SHBG has been explained.

The measurement of physiologically active testosterone in the blood is called the free androgen index (FAI). Most of the testosterone in the blood is either free or loosely bound to albumin, which is considered biologically active testosterone. The other part of testosterone that is strongly bound to SHBG is called the biologically inactive part. Just that fraction is deemed biologically active, whether it is free or linked to albumin. Therefore, one may determine the amount of physiologically active testosterone in the blood by measuring the ratio of total testosterone to SHBG [47]. The calculated FT and FAI, which mainly depend on TT and SHBG, showed a significant decrease in the AZA group, which may be due to hypogonadism [48]. The only treatment that achieved a significant increase in SHBG compared with the model group was the group supplemented with NS oil, indicating that NS oil had physiologically active testosterone in the blood. In a general view, there appears to be a positive relationship between FT and FAI from one side and TT and SHBG from the other with all experimental treatments, as previously reported [49].

FSH and LH/ICSH are two hormones involved in the endocrine stimulation of spermatogenesis [50], and it has been established that FSH and LH are essential for proper spermatogenesis. Gonadotropin-releasing hormone (GnRH) of the hypothalamus is responsible for balancing FSH and LH [51]. LH binds to its receptors on the surface of Leydig cells and stimulates the production of testosterone. FSH receptors are found in Sertoli cells, which also contribute to spermatogenesis regulation as well as the production of SHBG [51]. The significant reductions in serum FSH and LH with AZA administration recorded in the present study are compatible with the findings of previous studies [35]. The obtained findings for FSH and LH confirmed the previous studies that expose the animals to different types of stress [2,52]. All the experimental supplementations significantly upregulated levels of both FSH and LH even over those of the negative group, which was clear in the NS oil group. These findings are in line with previous studies [30,53] that concluded that TT, LH, and FSH were dramatically enhanced by Ns seed extractions. The beneficial effects of *Nigella sativa* extracts are owed to the boosting functioning of the hypothalamic–pituitary–testicular axis [54]. Honey supplementation during ischemia followed by reperfusion led to increased FSH, LH, and TT levels [50]. An amount of 200 mg/Kg of ethanolic extract of *Nigella sativa* enhanced TT and LH levels while having little impact on FSH levels [31]. Nevertheless, the reproductive hormones testosterone, LH, and FSH in rats were not significantly affected by Malaysian or Palestinian honey, according to other researchers [55]. This conflict may be attributed to the doses used of *Nigella sativa* and honey or due to the varieties of the floral origins of the honey, so further studies are required for more of an explanation.

Caspase-3 triggers the apoptotic process by causing the division of nuclear and cytosolic substrates, chromatin condensation, DNA fragmentation, and apoptotic bodies [56]. The obtained results display the apoptotic effect of AZA, which is represented in the significant elevations in the apoptotic markers caspase-3 and caspase-9 cascades compared with the negative control group, as previously recorded in [35], which reported upregulated levels of the apoptotic marker caspase-9. However, testicular caspase-3 showed upregulation with a monosodium glutamate stressor [38,57,58]. Likewise, high ROS caused damage to the mitochondrial membranes, triggered the release of cytochrome C, and accelerated the intrinsic apoptotic process in testicular tissue cells [57,59]. All the supplementations achieved the normalization of the caspase-9 cascades, which were emphasized in the NS oil group, whereas caspase-3 was mitigated with NS honey and NS oil supplementations but not with M-floral honey. However, the finding related to NS floral honey agrees with a previous study by [50], who concluded that pre-treatment with honey had a positive effect and decreased ROS, consequently leading to the inhibition of apoptosis. Similarly, chrysin (the main bioactive compound of honey) treatment against paracetamol or colistin toxicity mitigates caspase-3 activity [60].

NO normally regulates the functions of the male reproductive system, while under a pathologic state, it has adverse effects on steroidogenesis. NO causes caspase cascade activity and stimulates apoptosis by impairing the mitochondrial membranes of sperm [56]. Therefore, the present study considered NO as one of the apoptosis markers. The significant increase in NO recorded in the AZA model group is consistent with the previous work in [35], which found a significant increase in inducible NO synthase (responsible for NO production) and nuclear factor-ƘB-p65 (NF-ƘB-p65) expression in the AZA-treated group. All the treatments rebalanced NO concentration, suggesting that this was due to their antioxidants and anti-inflammatory contents. It is noticed that few studies that have been performed on NO, particularly in honey and NS, thereby corresponding antioxidants have been resorted to discussing the results. Antioxidants impair NO synthesis, resulting in a reduced production of NO [61]. The antioxidant activity of crispum can reduce NO and improve sperm quality [26]. Curcumin contains an antioxidant that can reduce NO concentration in varicocele male rats [62]. The antioxidant content of genistein could decrease NO through the inhibition of prostaglandins and reduction in nitric oxide synthesis [63].

Testicular tissue redox balance was represented in the current study by GSH-Px, SOD, CAT, and MDA, the products of lipid peroxidation. Enzymes that fight free radicals (ROS) are a crucial component of cellular defense. Oxidative stress may result from an imbalance between ROS generation and scavenging activities. Therefore, the redox balance in the body is interrupted and tissues may be damaged, leading to physiological dysfunction [64]. However, the significant reductions in GSH-Px and SOD, the marked decrease in CAT, and the significant elevation in MDA in testicular tissues recorded in this study in the AZA model group coincide with the previous work in [35]. NS floral honey and NS oil were able to rebalance the oxidant/antioxidant status in testicular tissue via the elevation in SOD and decrease in MDA. Moreover, NS oil achieved improvements in GSH-Px and CAT, suggesting that it has a potent antioxidative effect, which was confirmed by previous studies [12]. The antioxidant ability of NS oil might be owing to its content of the active ingredient thymoquinone, which has antioxidant activity and can protect tissue from oxidative injury [65]. The improvement in the redox balance exhibited in the NS floral honey group is consistent with other studies [66]. Honey is a rich source of antioxidant compounds that can restore redox balance by enhancing the activities of CAT and SOD [67]. Contrary to this, the improvement in the redox balance in the case of M-floral honey is not clear, except for an increase in SOD.

The sperm quality in the present study depended on the sperm count in three different parts of the epididymal (caudal, corpus, and caput), live sperm viability, and seminal fructose (as an indicator of sperm motility) due to seminal fructose, which is essential for spermatozoa metabolism and motility [68] and is considered a fuel for sperm motility [69]. In this study, the AZA model group demonstrated significant inhibition of the caudal epididymal sperm count (this finding was not recorded in the corpus and caput epididymides), sperm viability, and seminal fructose, and these findings are in line with another study [35]. NS oil was able to normalize the measured sperm quality and return its values to nearly equal to those of the negative control group. The beneficial role of *Nigella sativa* in sperm quality is reflected in another study [70] that demonstrated significant improvements in reproductive functions including sperm and Leydig cell counts. Likewise, in a clinical trial, daily use of NS oil (2.5 mL) by 68 infertile men for 8 weeks recorded an improvement in sperm morphology, count and motility, and semen volume [71]. NS oil contains several antioxidants including flavonoids, anthocyanin, carotene, isothiocyanate, and carotenoids, which can enhance sperm production and quality [72]. Similarly, the linoleic acid and oleic acid in NS oil can effectively improve sperm count, motility, and normality [73]. On the other hand, honey from two sources (M-floral honey and NS floral honey) did not influence sperm quality, and this finding agrees with the work of Dare et al. [74] [2013], who reported reduced motility and an increased percentage of dead and abnormal spermatozoa in honey-treated animals. Contrarily, the concomitant use of honey with toxic diethyl phthalate exposure has shown a significant improvement in healthy sperm count [3].

The histological examination of testicular tissues came in parallel to and confirmed the present data since the negative control group showed the typical histological architecture of seminiferous tubules with normal spermatogonia cells and complete spermatogenesis. This finding agrees with that described in the previous study by [35]. The AZA group showed a marked degeneration and necrosis of the spermatogonia cells lining seminiferous tubules, interstitial edema, and spermatid giant cells in the lumen. These histological demonstrations provide an idea about the effects of inflammation, which is associated with oxidative stress that impairs sperm quality, as reported by [75]. The drastic effect of AZA explains its germ cell apoptosis [76], as mentioned in this study. Honey supplementations from two floral sources (M-floral and NS floral) demonstrated the degeneration and necrosis of some spermatogonia cells lining some seminiferous tubules, the formation of spermatid giant cells in the lumen of the seminiferous tubule, and the necrosis of some Leydig cells, whereas few sections had the normal histopathology observed in the NS floral honey group, indicating that the supplementation with NS floral honey was better than that with M-floral honey to mitigate the inflammatory and oxidative effects of AZA on testicular tissue. However, when honey was supplemented with diethyl phthalate, a significant improvement in testicular histopathological lesions was shown [3]. Meanwhile, NS oil supplementation manifested a normal histological structure of seminiferous tubules, which is in line with [77]. All these histological observations confirm the obtained results of apoptosis, redox balance, and sperm quality previously mentioned in this study.

With an overall view of the present study, one can justify the benefits achieved by using NS oil for testicular degeneration, which may be attributed to its antioxidant contents, which can enhance steroid hormone synthesis, sperm production, and semen quality [72]. The antioxidant activity and anti-apoptotic properties of thymoquinone (the main constituent of black seed oil) can protect tissue from oxidative injury [65] and inhibit cell death, so it is essential to the integrity of mitochondria by releasing C cytochrome [78], which positively affects spermatogenesis and fertility [79]. In addition, NS oil contains linoleic acid and oleic acid, which have been reported to improve spermatogenesis [73]. The phenolic compounds in *Nigella sativa* play an important role in increasing the levels of follicular-stimulating hormone and testosterone in males [80]. On the other hand, the beneficial role of NS floral honey recorded in the present study may be attributed to its phenolic compounds of honey, which neutralize ROS, reduce lipid peroxidation, and increase the activities of antioxidant enzymes [81]. Chrysin (the main bioactive component of honey) acts as a testosterone-boosting agent and subsequently exhibits testosterone production [82]. Likewise, an antioxidant compound in honey, which is pantothenic acid, regulates the hormone for adrenal steroid secretion [83] and then works through hypothalamic hormone secretion to regulate the neuroendocrine–gonadal axis [84]. Furthermore, the antioxidant activity of honey may control anti-apoptotic patterns in testicular tissue [50].

## 5. Conclusions

With an overall view of the present results, it is notable that there is a direct and strong relationship between the obtained findings of the pituitary–testicular hormone axis, apoptosis cascade, redox balance from one side, and spermatogenesis and fertility from the other side. NS oil supplementation restored the functional integrity of the pituitary–gonadal axis, TT synthesis, and sperm production and quality. The honey supplementations from two various floral sources differ in their effects in attenuating the gonadotoxicity caused by the administration of AZA. The M-floral honey group did not show clear and reliable results in most of the measurements owing to the non-specific floral origin that might contain poor bioactive compounds, phenols, and antioxidants. We claim that it is necessary to conclusively take into account the floral origin of honey. Unexpectedly, despite NS floral honey’s success in achieving a beneficial effect to protect against testicular damage, it did not upgrade to the level of improvement recorded in the NS oil group. Further clinical studies are essential to detect the qualities and quantities of the bioactive constituents of NS floral honey to ensure its use as part of a daily routine for persons who are exposed to pollution, chemotherapy, or any substances that produce oxidative stress.

## 6. Research Limitation

For more transparency, this research has a limitation, as it relied on previously published data on the bioactive compounds and phytochemical constituents of *Nigella sativa* and honey to explain the benefits obtained from their supplementations. The authors think that the active ingredients and the phenolic compounds in *Nigella sativa* do not change much and are established and well-known. Nevertheless, the authors recommend essential further clinical studies to focus on the qualities and quantities of these bioactive constituents.

## Figures and Tables

**Figure 1 nutrients-15-01693-f001:**
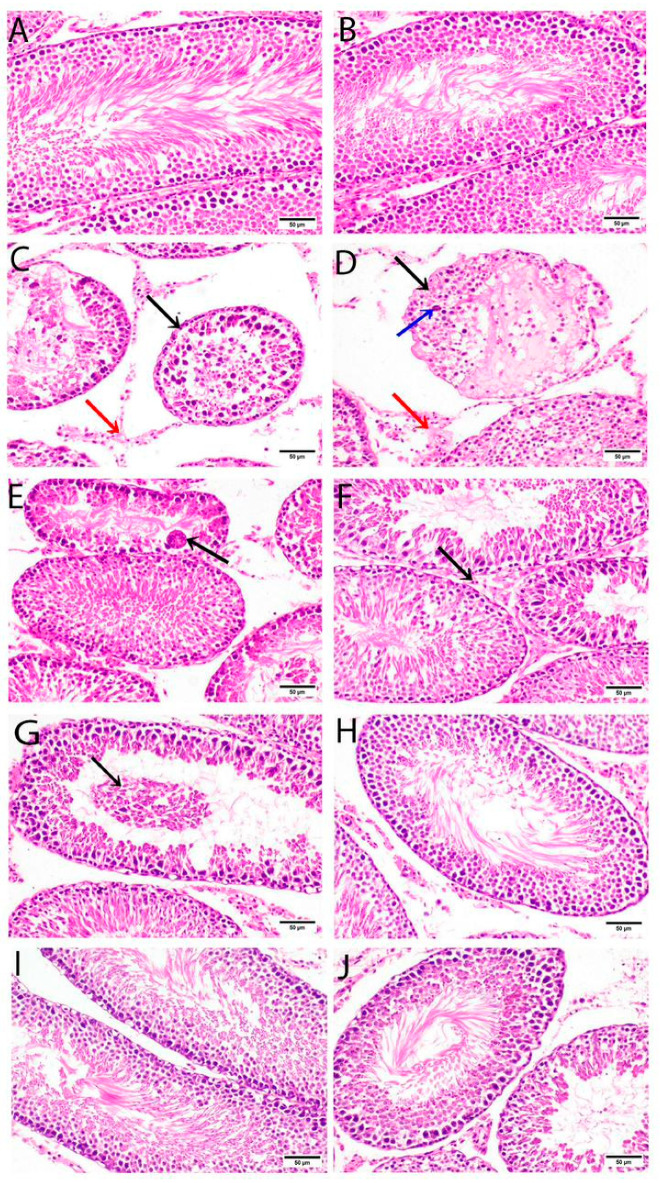
Testicular sections from all experimental groups were stained with hematoxylin and eosin (H&E; ×400; scale bar, 50 μm). The testes of negative control rats (**A**,**B**) show a typical histological structure of seminiferous tubules with normal spermatogonia cells and complete spermatogenesis. Testes of positive control model rats administered AZA revealed marked degeneration and necrosis of spermatogonia cells lining seminiferous tubules (black arrow) and interstitial edema (red arrow) (**C**) and marked degeneration and necrosis of spermatogonia cells lining seminiferous tubules (black arrow) associated with spermatid giant cell in the lumen (blue arrow) and interstitial edema (red arrow) (**D**). Testes of rats administered M-floral honey revealed the formation of spermatid giant cells in the lumen of the seminiferous tubule (arrow) (**E**) and necrosis of some Leydig cells (arrow) (**F**). Testes of rats administered NS floral honey demonstrated necrosis and sloughing of some spermatogonia cells lining some seminiferous tubules (arrow) (**G**), whereas other sections revealed no histopathological alterations (**H**). The testes of rats administered NS oil manifested the typical histological structure of seminiferous tubules (**I**,**J**).

**Table 1 nutrients-15-01693-t001:** Effects of M-floral honey, NS floral honey, and NS oil on body gain and testis weight.

	Parameter	Initial Body Weight (g)	Final Body Weight (g)	Body Weight Gain (g)	Testis Weight (g)	Relative Testis Weight (%)
Group						
Negative control		156.6 ± 3.5	276.2 ± 6.2	119.8 ± 3.7	1.72 ± 0.18	0.624 ± 0.071
AZA model		160.3 ± 2.7	237.3 ± 7.8 *	77.4 ± 8.5 **	0.96 ± 0.05 *	0.408 ± 0.012 *
M-floral honey		159.8 ± 3.4	252.9 ± 4.6	93.7 ± 4.5	1.51 ± 0.25	0.605 ± 0.047 ^a^
NS floral honey		155.7 ± 2.7	271.8 ± 5.0 ^a^	116.4 ± 5.7 ^a^	1.53 ± 0.12 ^a^	0.566 ± 0.042 ^a^
NS oil		154.9 ± 3.6	269.2 ± 4.2 ^a^	114.5 ± 3.9 ^a^	1.67 ± 0.16 ^a^	0.624 ± 0.060 ^a^
*p*-value		<0.1364	<0.0437	<0.0333	<0.01543	<0.03641

Means ± standard error (SE). Values in the same column with the marks * and ** of the AZA model group differ significantly from the value of the healthy control group at *p* < 0.05 and *p* < 0.01, respectively. Values of the treated groups with the letter a differ significantly from the value of the AZA model group at *p* < 0.05. AZA: azathioprine; M-floral honey: multi-floral honey; NS floral honey: *Nigella sativa* floral honey; and NS oil: *Nigella sativa* oil.

**Table 2 nutrients-15-01693-t002:** Effects of M-floral honey, NS floral honey, and NS oil on testosterone hormone, SHBG, and related parameters.

	Parameter	TT (ng/mL)	SHBG (ng/mL)	Free Testosterone (FT)	Free Androgen Index (FAI)	TT/FT Ratio
Group						
Negative control		5.35 ± 0.77	3.27 ± 0.27	2.10 ± 0.38	163.53 ± 4.31	2.57 ± 0.45
AZA model		2.61 ± 0.41 *	1.84 ± 0.18 **	0.78 ± 0.09 *	141.82 ± 4.28 *	3.21 ± 0.79
M-floral honey		3.87 ± 0.95	2.43 ± 0.53	1.47 ± 0.34	159.31 ± 4.64	2.69 ± 0.34
NS floral honey		4.97 ± 0.42 ^a^	3.16 ± 0.37 ^a^	1.82 ± 0.29	157.77 ± 4.93	2.76 ± 0.61
NS oil		4.82 ± 0.38 ^a^	2.97 ± 0.23 ^a^	1.87 ± 0.17 ^b^	162.11 ± 3.74 ^a^	2.60 ± 0.37
*p*-value		<0.00232	<0.001240	<0.03332	<0.02931	<0.06482

Means ± standard error (SE). Values in the same column with the marks * and ** of the AZA model group differ significantly from the value of the healthy control group at *p* < 0.05 and *p* < 0.01, respectively. Values of the treated groups with the letters a and b differ significantly from the value of the AZA model group at *p* < 0.05 and *p* < 0.01, respectively. AZA: azathioprine; M-floral honey: multi-floral honey; NS floral honey: *Nigella sativa* floral honey; and NS oil: *Nigella sativa* oil.

**Table 3 nutrients-15-01693-t003:** Effects of M-floral honey, NS floral honey, and NS oil on gonadotrophins.

	Parameter	Gametogenic H (FSH) (ng/mL)	ICSH (LH) (mlU/mL)
Group			
Negative control		4.56 ± 0.11	8.78 ± 0.56
AZA model		4.11 ± 0.07 *	6.23 ± 0.49 *
M-floral honey		4.77 ± 0.19 ^a^	8.27 ± 1.34 ^a^
NS floral honey		4.73 ± 0.14 ^b^	8.84 ± 1.22 ^a^
NS oil		4.85 ± 0.21 ^a^	9.62 ± 1.10 ^a^
*p*-value		<0.00241	<0.00363

Means ± standard error (SE). Values in the same column with the mark * of the AZA model group differ significantly from the value of the healthy control group at *p* < 0.05. Values of the treated groups with the letters a and b differ significantly from the value of the AZA model group at *p* < 0.05 and *p* < 0.01, respectively. AZA: azathioprine; M-floral honey: multi-floral honey; NS floral honey: *Nigella sativa* floral honey; and NS oil: *Nigella sativa* oil.

**Table 4 nutrients-15-01693-t004:** Effects of M-floral honey, NS floral honey, and NS oil on caspase-3, caspase-9, and nitric oxide.

	Parameter	Caspase-3 (ng/mg Protein)	Caspase-9 (ng/mg Protein)	Nitric Oxide (µmol/mg Protein)
Group				
Negative control		10.32 ± 1.43	1.70 ± 0.17	0.064 ± 0.008
AZA model		25.63 ± 2.54 *	2.83 ± 0.13 **	0.141 ± 0.014 *
M-floral honey		20.81 ± 3.11	1.76 ± 0.26 ^a^	0.068 ± 0.012 ^b^
NS floral honey		14.23 ± 2.08 ^a^	1.83 ± 0.29 ^a^	0.064 ± 0.017 ^a^
NS oil		13.40 ± 3.81 ^a^	1.60 ± 0.18 ^b^	0.058 ± 0.016 ^a^
*p*-value		<0.01117	<0.03442	<0.01432

Means ± standard error (SE). Values in the same column with the marks * and ** of the AZA model group differ significantly from the value of the healthy control group at *p* < 0.05 and *p* < 0.01, respectively. Values of the treated groups with the letters a and b differ significantly from the value of the AZA model group at *p* < 0.05 and *p* < 0.01, respectively. AZA: azathioprine; M-floral honey: multi-floral honey; NS floral honey: *Nigella sativa* floral honey; and NS oil: *Nigella sativa* oil.

**Table 5 nutrients-15-01693-t005:** Effects of M-floral honey, NS floral honey, and NS oil on redox balance.

	Parameter	GSH-Px (U/mg Protein)	SOD (U/mg Protein)	CAT (U/mg Protein)	MDA (nmol/g Protein)
Group					
Negative control		11.42 ± 1.04	20.83 ± 2.5	5.71 ± 1.34	25.8 ± 4.2
AZA model		5.43 ± 1.28 *	11.61 ± 1.2 *	2.84 ± 0.36	45.7 ± 3.7 *
M-floral honey		8.14 ± 1.95	17.53 ± 3.2	6.17 ± 1.24	30.8 ± 4.8
NS floral honey		5.50 ± 1.87	19.76 ± 1.9 ^a^	6.53 ± 1.93	26.2 ± 3.1 ^a^
NS oil		10.46 ± 0.41 ^a^	22.85 ± 2.6 ^a^	6.83 ± 1.23 ^a^	24.8 ± 4.3 ^a^
*p*-value		<0.03215	<0.02863	<0.01749	<0.02953

Means ± standard error (SE). Values in the same column with the mark * of the AZA model group differ significantly from the value of the healthy control group at *p* < 0.05. Values of the treated groups with the letter a differ significantly from the value of the AZA model group at *p* < 0.05. AZA: azathioprine; M-floral honey: multi-floral honey; NS floral honey: *Nigella sativa* floral honey; and NS oil: *Nigella sativa* oil.

**Table 6 nutrients-15-01693-t006:** Effects of M-floral honey, NS floral honey, and NS oil on sperm count, sperm viability, and fructose.

	Parameter	Epididymal Sperm Count (10^6^/mL)			Sperm Viability (%)	Fructose (mmol/L)
Group		Caudal	Corpus	Caput		
Negative control		92.56 ± 4.51	48.74 ± 2.8	38.74 ± 2.3	88.56 ± 4.5	13.57 ± 1.2
AZA model		54.59 ± 9.94 **	38.66 ± 3.6	32.26 ± 1.1	63.43 ± 4.1 *	22.23 ± 2.02 *
M-floral honey		68.43 ± 4.61	42.34 ± 3.1	36.21 ± 2.8	77.38 ± 3.7	18.81 ± 1.8
NS floral honey		63.12 ± 5.53	45.11 ± 2.5	37.38 ± 3.1	74.28 ± 2.6	17.98 ± 2.3
NS oil		85.87 ± 3.42 ^a^	44.09 ± 4.1	34.73 ± 2.4	83.73 ± 2.8 ^a^	14.91 ± 1.4 ^a^
*p*-value		<0.00321	<0.15468	<0.11112	<0.04333	<0.00763

Means ± standard error (SE). Values in the same column with the marks * and ** of the model group differ significantly from the value of the healthy control group at *p* < 0.05 and *p* < 0.01, respectively. Values of the treated groups with the letter a differ significantly from the value of the model group at *p* < 0.05. AZA: azathioprine; M-floral honey: multi-floral honey; NS floral honey: *Nigella sativa* floral honey; and NS oil: *Nigella sativa oil.*

## Data Availability

The corresponding author can provide all the data used in the present study upon reasonable request.

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
