# Peer review of "Nigella sativa-Floral Honey and Multi-Floral Honey versus Nigella sativa Oil against Testicular Degeneration Rat Model: The Possible Protective Mechanisms"

_nutrients, 2023, doi:10.3390/nu15071693_

Round 1
Reviewer 1 Report
Infertility is a global health issue affecting millions of people of reproductive age worldwide. The male reproductive function particularly, the testes and related hormones are sensitive to various xenobiotics. This work aimed for the first time to assess Nigella sativa-floral honey and multi-floral honey versus Nigella sativa oil against rat testicular degeneration induced by azathioprine. The authors found remarkable results, they found out that a direct relationship exists between the pituitary-testicular hormone axis, apoptosis cascade, redox balance from one side, spermatogenesis, and fertility from the other side. Based on the results, after the necessary clinical studies to determine the quality and quantity of bioactive components, NS-flower honey may have a prospective use in the targeted therapy/treatment of infertility caused by oxidative stress.
I have no additional remarks or comments regarding the article. After consideration, I suggest to the editor that this manuscript is accepted for publication in Nutrients in present form.
Author Response
Dear reviewer(s),
We appreciate the time devoted to looking at our manuscript. The revised version of the manuscript has been modified according to the academic editor comments.
Reviewer 2 Report
The authors submitted an original research interesting paper, which deals with protective effects of various parts or products Nigella saliva species, such as honey or oil, against oxidative stress in testes of experimental animals. Since oxidative damages of tests are often reasons of male infertility, the topic of the manuscript is important and timely.
The authors investigate several aspects of the antioxidant protection of the plant species products. They analysed redox status of the tissues, apoptosis markers, and additionally the authors explored the histopathological markers. All of the used methods were adequate. The results are illustratively presented and properly discussed.
Author Response

(The authors gave the same response as above.)
